# Two Centuries of Drought History in the Center of Chihuahua, Mexico

**Aldo Rafael Martínez-Sifuentes [1], José Villanueva-Díaz [1], Juan Estrada-Ávalos [1], Ramón Trucíos-Caciano [1,\*], Teodoro Carlón-Allende [2] and Luis Ubaldo Castruita-Esparza [3]**

[1] Instituto Nacional de Investigaciones Forestales, Agrícolas y Pecuarias, Centro Nacional de Investigación Disciplinaria en Relación Agua, Suelo, Planta y Atmósfera. Km. 6.5 Margen Derecha Canal de Sacramento, Gómez Palacio 35150, Mexico; martinez.aldo@inifap.gob.mx (A.R.M.-S.); villanueva.jose@inifap.gob.mx (J.V.-D.); estrada.juan@inifap.gob.mx (J.E.-Á.)

[2] CONACYT-Instituto de Geofísica, Unidad Michoacán, Universidad Nacional Autónoma de México, Antigua Carretera a Pátzcuaro No. 8701, Col. Ex-Hacienda de San José de la Huerta, Morelia 58190, Mexico; tcarlon@igeofisica.unam.mx

[3] Facultad de Ciencias Agrícolas y Forestales, Universidad Autónoma de Chihuahua, Km. 2.5 Carretera Delicias a Rosales, Delicias 33000, Mexico; lcastruita@uach.mx

\* Correspondence: trucios.ramon@inifap.gob.mx

**Abstract:** Droughts are a climatic phenomenon with local, regional, and large-scale repercussions. Historical knowledge of droughts generated by modeled data allows the development of more accurate climate reconstructions to propose better approaches for the management of hydric resources. The objective of this research was to evaluate the association of precipitation and temperature with data from the NLDAS-002 to develop a reconstruction of droughts in the center of Chihuahua, Mexico using the SPEI from tree rings. We also identified the influence of ocean–atmospheric phenomena on the reconstructed drought index. The best association among chronologies was obtained with the earlywood band and accumulated seasonal precipitation from November of the previous year to June of the current year (r = 0.82, $p < 0.05$) and for temperature from January to July (r = −0.81, $p < 0.05$). The reconstructed drought index extended from 1775 to 2017 (243 years), where seven extreme drought events were identified. We found significant correlations between the reconstructed Standardized Precipitation Evapotranspiration Index and the Pacific Decadal Oscillation (r = 0.46, $p < 0.05$), Atlantic Multidecadal Oscillation (r = −0.34, $p < 0.05$), Multivariate El Niño Southern Oscillation Index (r = 0.29, $p < 0.05$), and Southern Oscillation Index (r = −0.22, $p < 0.05$). The historical reconstruction of hydroclimatology in the center of Chihuahua is important for planning a long-term assessment and for the management of water resources shared by Mexico and the United States.

**Keywords:** precipitation; drought index; tree rings; temperature

## 1. Introduction

Extreme hydrological events such as maximum floods and extreme droughts of high intensity and duration have negative economic and ecological effects on society [1]. Droughts are a high-priority problem in terms of the conservation of biodiversity and human survival on a global scale [2]. It is difficult to define a drought because they are spatially and temporally variable, and may have different impacts at different landscape scales [3]. In arid areas of Mexico, the increase in temperature, the decrease in precipitation and the concomitant demand for water resources for urban, agricultural, and natural resources heightens the need for more drought studies [4].

Droughts occur at multiple time scales and originate from water deficit in the natural climatic process and they have an impact on productive systems, such as industrial and agricultural activities [5]. Droughts based on their impacts are categorized into four

types—meteorological, agricultural, hydrological, and socio-economical [6]. In this context, evapotranspiration is a factor influenced by different variables, including temperature and atmospheric evaporation [7]. Temperature is the climatic variable that directly influences evapotranspiration and, therefore, the severity of the drought [8]. Several drought indices were developed; among the most common are the Palmer Drought Severity Index (PDSI) [9], and the Standardized Precipitation Index (SPI) [10]. The PDSI is based on a hydrological balance that considers parameters such as precipitation, temperature, and a soil moisture factor, in comparison to the SPI, which only involves precipitation as input [4]. Another index widely used in drought studies is the Standardized Precipitation Evapotranspiration Index (SPEI) [4].

The SPEI is a multiscalar index that takes into account current and historical precipitation and evapotranspiration [11] that identifies droughts at a different time scale based on the trends produced by climate change in a region [12]. Droughts, although common in the semi-arid region of Mexico in recent decades, have affected the central-northern region of the country with greater intensity and frequency [13]. Information on the social and economic impact of the most recent droughts is widely documented; the drought that occurred from 1987 to 1989 resulted in decreased electrical energy from lost hydrological power at the dam, water availability, and agricultural production, with an estimated cost of MXN 39 billion [14].

Northern Mexico is a region prone to frequent droughts, due to dominant arid conditions of its orography, in addition to being located in the subtropical high-pressure belt (STHP) [1,15]. The center of Chihuahua is characterized by a series of environmental problems with international impacts, such as land-use change, over-extraction of minerals from the riverbed, water pollution, and frequent droughts [16]. The region is influenced by the teleconnection of the El Niño Southern Oscillation phenomenon (ENSO) [17]; particularly, the warm phase El Niño, which causes winter rains and above-average runoff [18], whereas the cold phase La Niña causes extreme droughts [19].

Northern Mexico in general has been the subject of numerous climate studies. Tae-Woong et al. [20] developed an analysis to identify severity events and return periods of droughts using the PDSI with time restriction due to the PDSI data extension. Villanueva-Díaz et al. [21] developed precipitation reconstructions for northwestern Chihuahua, southeastern Chihuahua, and Basaseachi National Park in the Western Sierra Madre, respectively. Those reconstructions have important implications in planning and water resource use in irrigation districts such as the Yaqui River in Sonora, which depends on water resources in the upper reaches of the mountain range. Woodhouse et al. [18] developed precipitation and streamflow reconstructions in the Conchos River basin based on six chronologies distributed in the center and northern Chihuahua. One of their observations was the need to expand the network of tree ring chronologies in the basin for a better understanding of the historical climate variability and water yield for planning purposes. Thus, it is important to generate multiscale studies to characterize the variability of the droughts through an actual and robust dendrochronological network considering drought indicators, such as the SPEI. Based on observed data and land surface modeling, databases were generated by international institutions and are now an available source of climate data for North America. These sources of information offer opportunities to obtain spatially and temporally relevant data that describes large-scale interactions of climate dynamics and cycles that regulate local and regional conditions [22]. The North American Land Data Assimilation System v002 (NLDAS-2) is a reconstructed modeled dataset, which provides relevant information for the determination of water balances, drought incidents, precipitation-runoff ratios, and other associated parameters [23]. Regarding physiological response, dendroclimatic studies were developed using diverse conifer genera in Mexico [24]. For species such as *Picea chihuahuana* Martínez, earlywood growth depends on the December–March winter rainfall, while latewood only for January [25]. In *Pinus cembroides* Zucc., earlywood is significantly related to rainfall from January to July [26]. The earlywood of *Pinus arizonica* Engelm depends on rainfall from October to May and the latewood in the same period but

weaker [27]. Seasonal rainfall in January–May presents a strong influence on earlywood growth of *Pseudotsuga menziesii* (Mirb.) Franco in northern Mexico [28], similar to the behavior of *Pinus douglasiana* Martínez with a response to the same period for earlywood growth [29]. The influence of climate on the growth of *Pinus lumhiltzii* Robins&Ferns begins with rainfall in the previous year for the formation of the earlywood ring until the middle of the current year [30]. The objectives of the present research were to: (1) generate a regional climatic association of precipitation and temperature with data from the NLDAS-2, (2) generate a historical reconstruction of the SPEI in central Chihuahua, (3) determine extreme drought events, and (4) identify the influence of oceanic–atmospheric phenomena with a reconstructed SPEI.

## 2. Materials and Methods

### 2.1. Study Area and Dendrochronological Data

The present study was developed in central Chihuahua, Mexico (Figure 1), located in the extreme coordinates 26°05′ to 29°55′ N, and 104°20′ to 107°55′ W, with elevation from 772 to 3282 masl. Dominant climate in the basin varies from very arid, arid, semi-arid, and sub-humid [31]. Annual average temperature fluctuates from 8 °C to 18 °C with an average annual precipitation of 419 mm [32] (Figure 2).

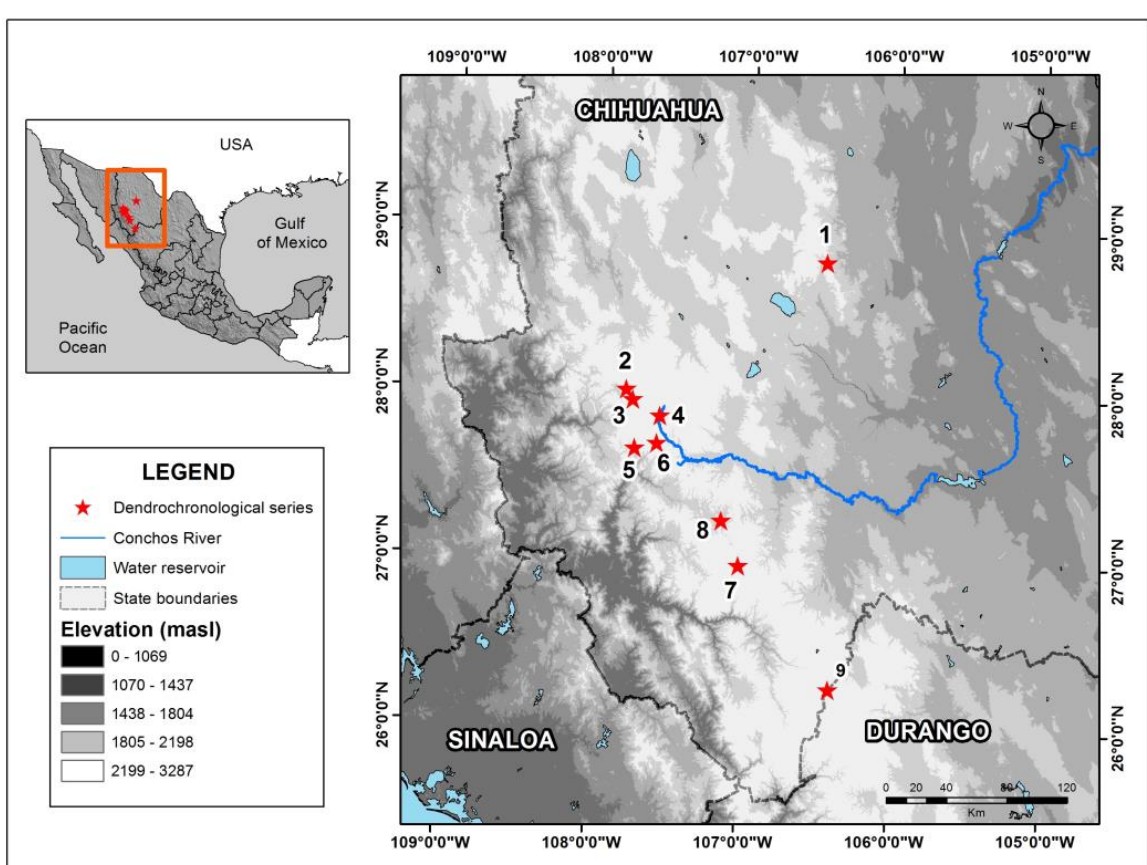

**Figure 1.** Geographic location of the dendrochronological series in central Chihuahua.

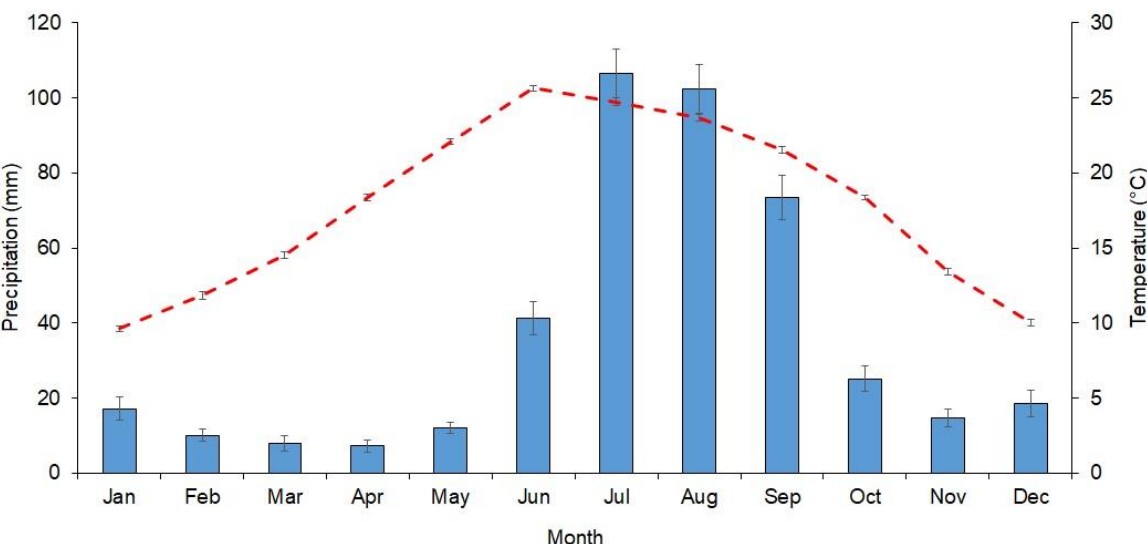

**Figure 2.** Climogram of the study area.

A total of 27 earlywood, latewood, and total ring-width chronologies from *Pinus cembroides*, *Pinus arizonica*, *Pinus durangensis*, *Pinus lumholtzii*, and *Pseudotsuga menziesii* were downloaded from the International Tree Ring Database [33] (Table 1, Figure 1). These dendrochronological series were developed in the Laboratory of Dendrochronology of the Research Institute of Forestry, Agriculture, and Livestock in Mexico and made available to the international tree ring database portal (https://www.ncdc.noaa.gov/data-access/paleoclimatology-data/dataset/tree-ring, accessed on 10 November 2021).

**Table 1.** Tree ring chronologies analyzed in central Chihuahua.

| Number | Site | Site Code | Extension (Years) | Species [1] | Series Intercorrelation | Chronology Type [2] |
|---|---|---|---|---|---|---|
| 1 | Majalca | MAJ | 1750–2013 (264) | Pce | 0.65 | RW, EW, LW |
| 2 | Basagochi | CAC | 1809–2013 (205) | Pme | 0.69 | RW, EW, LW |
| 3 | Ranchito San Juanito | RAN | 1770–2013 (244) | Pch | 0.54 | RW, EW, LW |
| 4 | Baburiachi | BAB | 1889–2012 (124) | Par | 0.60 | RW, EW, LW |
| 5 | Barranca del Cobre | COB | 1745–2014 (270) | Pme | 0.64 | RW, EW, LW |
| 6 | Arareco | ARA | 1874–2014 (141) | Par | 0.50 | RW, EW, LW |
| 7 | El Tule Gpe. Y Calvo | ELT | 1830–2013 (184) | Pdu | 0.54 | RW, EW, LW |
| 8 | Guachochi | GUA | 1806–2017 (212) | Plu | 0.62 | RW, EW, LW |
| 9 | Los Pilares | LPI | 1725–2015 (291) | Pme | 0.69 | RW, EW, LW |

[1] Pce = *Pinus cembroides*, Pme = *Pseudotsuga menziesii*, Pch = *Picea chihuahuana*, Par = *Pinus arizonica*, Pdu = *Pinus durangensis*, Plu = *Pinus lumholtzii*. [2] RW = Total Ring Width, EW = Earlywood, LW = Latewood.

The chronologies were developed by conventional dendrochronological techniques, which involve the dating and measurement of each tree ring using a Velmex system with a precision of 0.001 mm; COFECHA software was used for quality control, considering an intercorrelation of 0.328 ($p < 0.01$) between series for an appropriate dating [34]. The standardized data were generated through negative exponential curves and straight lines of a positive or negative slope with ARSTAN software, which minimizes biological effects caused by radial increase with age and disturbances of natural or anthropogenic origin [35]. The indicator of correlation between the variance of the chronology with the theoretical population (i.e., Expressed Population Signal (EPS)) was calculated with R [36]. Mérian et al. [37] suggested that EPS value should be ≥0.85. To maximize the climate signal,

a regional chronology was generated by combining all the individual total ring-width series to create a regional total-ring width chronology, a process developed for both earlywood and latewood chronologies.

### 2.2. Climatic Variables

Data were obtained from the NLDAS-2 database (https://disc.gsfc.nasa.gov/datasets/NLDAS_FORA0125_M_002/summary?keywords=NLDAS accessed on 25 November 2021). The database stores climatic information generated from the integration of observed data and modeling of the Earth's surface [38]. The information includes accumulated monthly and hourly climate data from 1979 to date, with a spatial resolution of $0.125° \times 0.125°$ in latitude–longitude.

### 2.3. Standardized Precipitation–Evapotranspiration Index

The SPEI values were downloaded from the open-access global drought monitor (http://spei.csic.es/map/maps.html#months=1#month=3#year=2018 accessed on 5 December 2021); we worked on a three-month time scale because this resolution establishes an association at the seasonal and annual level [39]. The downloaded data extend from 1951 to 2017, and consist of the average value of SPEI for the gridded data according to the study area with a spatial resolution of $0.5° \times 0.5°$ in latitude–longitude.

### 2.4. Statistical Analysis

The association between the regional chronology and climatic variables, and between reconstructed SPEI and climatic variables, were developed using the DENCROCLIM software [40] by seasonal, accumulated, and annual windows (40 years). Maps with the best correlations were developed to obtain a spatial and temporal visualization of the association between the growth of the tree species that composed the regional chronology and the monthly average temperature and precipitation, using the KNMI climate explorer [41]. For the period with the highest significant correlation ($p < 0.05$) between the regional chronology and the SPEI, a regression equation was generated and used as a transfer function, which was validated with sub-periods generated by half of the records observed and half of the reconstructed data using the verify subroutine from the Dendrochronology Program Library [42]. We used Minitab v17 software to assess the transfer model, and to calculate explained variance ($R^2$), reduction of error, and coefficient of efficiency for the two sub-periods.

The extreme periods of reconstructed SPEI were calculated by using the values outside the 95th percentile. The spatial and temporal relationship between regional chronology and SPEI with $0.5°$ spatial resolution was explored by spatial correlation analysis through the KNMI climate explorer [40]. The spectral frequencies and significant period of drought based on SPEI values were determined through a Multi-taper method of spectral analysis [43].

### 2.5. Influence between Ocean–Atmosphere Phenomena and Reconstructed SPEI

For the analysis, the reconstructed SPEI was compared using a similar period as the ENSO, through the Southern Oscillation Index (SOI) [17] and Multivariate ENSO Index (MEI) [44], as well as with the Atlantic Multidecadal Oscillation (AMO) [45], and the Pacific Decadal Oscillation (PDO) [46].

The comparison of the reconstructed SPEI to the reconstructed June, July, August Palmer Drought Severity Index [24] was determined by Pearson correlation analysis with STATISTICA 8 software.

## 3. Results

### 3.1. Tree Ring Chronologies and Association between Climatic Variables

The regional chronology extends from 1725 to 2017 (Figure 3a) but was reduced to the period 1775–2017 (Figure 3b) where an EPS > 0.85 has nine cores. The total ring-width

and latewood dendrochronological series were discarded from further analysis due to their lower response to climatic variables as compared with the standard version of the earlywood chronology.

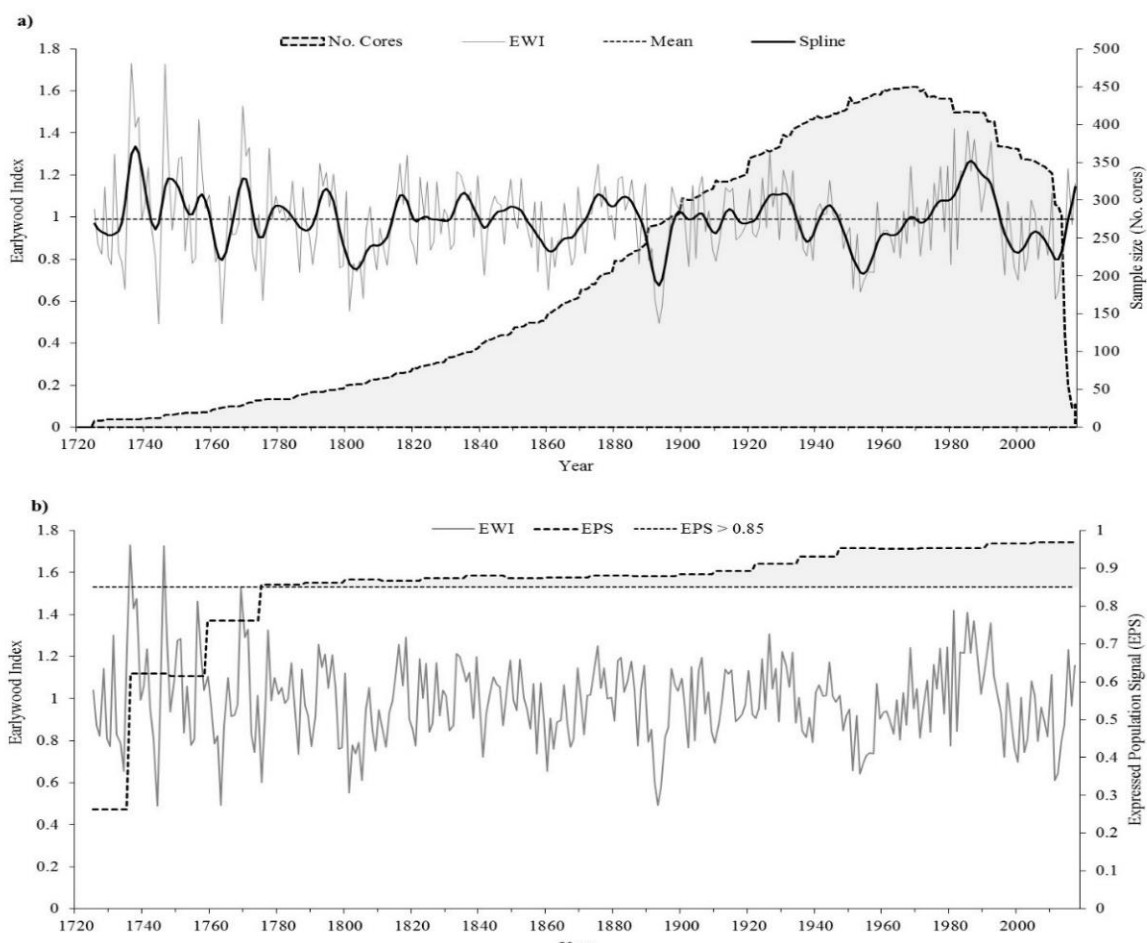

**Figure 3.** (**a**) Regional earlywood chronology (gray line), with a ten-year flexible spline to highlight low-frequency events (black line); the dotted black line represents the number of cores. (**b**) Regional earlywood chronology standard version (gray line); the EPS is represented by a dotted black line, and EPS > 0.85 limit is represented by a broken horizontal line in gray.

Pearson's correlation analysis between the earlywood chronology and monthly precipitation showed several significant periods ($p < 0.05$). Among them, the seasonal period January–July stands out, with correlations from r = 0.28 for July to r = 0.67 for January of the current growth year, and a highly significant correlation (r = 0.82) for the cumulative seasonal period (November of the previous year to June of the current growth year) (Figure 4a).

The correlations with monthly temperature were mostly negative both in the current year and in the previous year; significant negative correlations were found in January (r = −0.63), March (r = −0.50), May (r = −0.53), and June (r = −0.60). Regarding the previous year, all values were significant, except December, although the period with significant correlation was obtained with the average seasonal period from January to July of the current growth year (r = −0.81) (Figure 4a).

The spatial correlation between the regional chronology and average monthly precipitation from November of the previous year to June of the current growth year shows a significant association (r > 0.6, $p < 0.05$) for central and northern Mexico (Figure S1a). Likewise, the average monthly temperature from January to July of the current growth

year presented a significant but negative association (r > −0.6, *p* < 0.05) with the regional chronology in central and northern Mexico (Figure S1b).

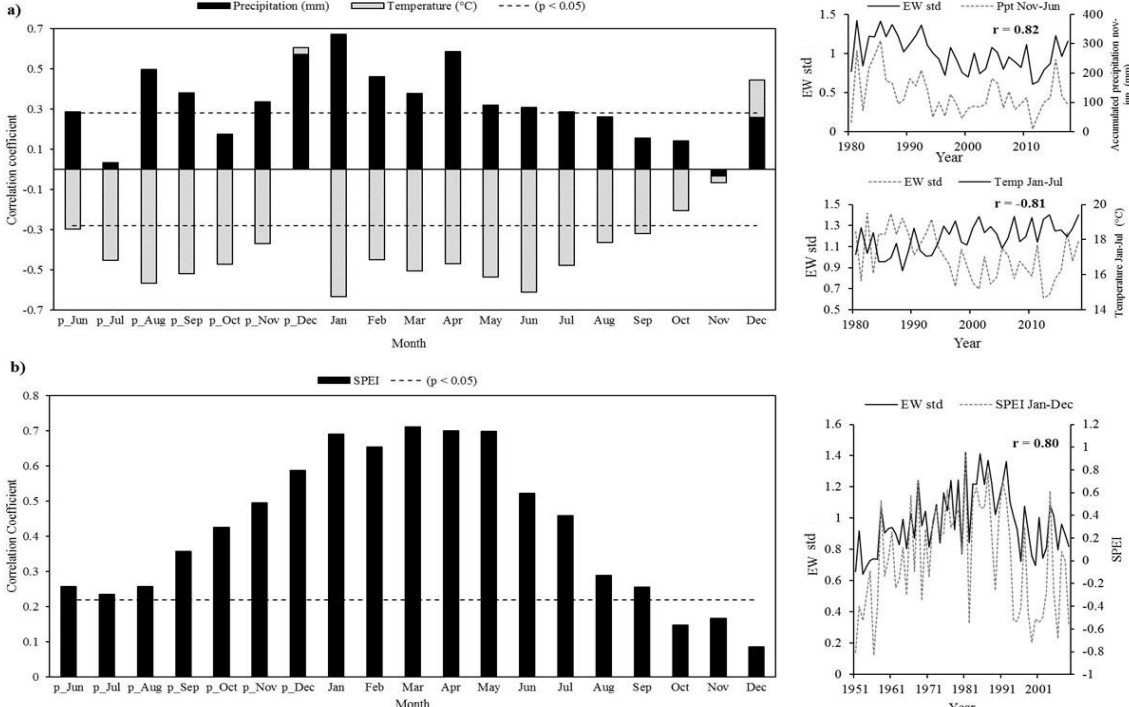

**Figure 4.** (**a**) Correlation between regional chronology and monthly mean precipitation (black bars) and monthly mean temperature (gray bars). On the right side, the best association with earlywood indices is presented for accumulated precipitation in November of the previous year to June of the current year (upper right figure), and with mean January-July temperature of the current year; (**b**) Correlation between regional chronology and monthly SPEI. The dotted horizontal lines indicate the level of significance (*p* < 0.05). On the right side, the association between the annual SPEI and the earlywood of the regional chronology is presented.

### 3.2. Association between the Regional Tree Ring Chronology and SPEI

The association between indices of the regional earlywood chronology and the SPEI showed significant correlations (*p* < 0.05) in most of the analyzed months, except October, November, and December of the current growth year. The highest correlation (r = 0.71) was presented in March, followed by April and May (r = 0.69) (Figure 4b). The January–December period presented a correlation of 0.80 and was used for the development of the SPEI reconstruction, which expresses the annual behavior of droughts in central Chihuahua.

The correlation between the regional chronology and the annual SPEI through spatial exploration (Figure S2) allowed us to delimit areas where the environmental conditions had a significant influence (*p* < 0.05) on the growth of conifers in north-central Mexico. The spatial representation of this association showed a region with significant influence (r > 0.6) in northern Chihuahua and Coahuila, in Mexico, as well as in New Mexico and Texas, in the United States of America.

### 3.3. Reconstruction of SPEI and Multi-Taper Method of Spectral Analysis

The SPEI was reconstructed from January to December based on the correlation and regression analysis between the earlywood regional series and the SPEI. The linear model for the period 1951–2009 is:

$$Y = 1.8615 \text{ x* } X - 1.8069 \tag{1}$$

where Y is the reconstructed annual SPEI value, and X is the earlywood index. The earlywood chronology explains 64% of the variance of the annual SPEI. The calibration-

verification tests on half of the instrumental records presented adequate validation for reconstruction effects (Table 2).

**Table 2.** Validation statistics of the SPEI annual reconstruction model.

| | Period | |
| Statistic | Calibration (1951–1980) | Verification (1981–2009) |
|---|---|---|
| Explained variance | 0.62 * | 0.77 * |
| Reduction of error | 0.54 * | 0.70 * |
| *t*-value | 3.47 * | 5.78 * |
| Signs test | 8 * | 5 * |
| First negative difference | 6 * | 7 * |
| Efficiency coefficient | 0.54 | 0.70 |

* Significant ($p < 0.05$).

The annual reconstruction of the SPEI extends from 1775 to 2017 (243 years). The results indicate the presence of extreme drought events (outside the 95th percentile) in 1775, 1801, 1805, 1860, 1892-1894, 1951, 1953-1954, 2000, and 2011-2012. A 10-year flexible spline was fitted to highlight low-frequency events (Figure 5).

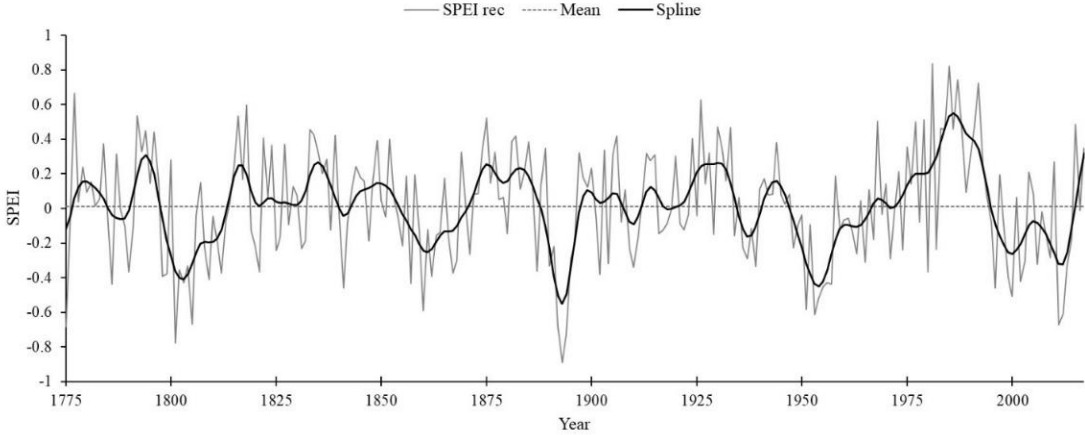

**Figure 5.** SPEI reconstructed from 1775 to 2017 (243) years for central Chihuahua.

The multi-taper spectral analysis of the reconstructed SPEI shows significant cycles at 55 years ($p < 0.01$) and every 2.1 years ($p < 0.05$); these frequencies are presented on an interannual and multidecadal scale (Figure S3a). According to the wavelet analysis, these frequencies were significant from 1860 to 1900 and from 1925 to 1950 ($p < 0.05$) (Figure S3b).

*3.4. Influence between Ocean–Atmosphere Phenomena and Reconstructed SPEI*

The PDO is the oceanic–atmospheric phenomenon with higher significant influence on the environmental conditions inferred through SPEI (r = 0.46, $p < 0.05$), and significantly but in anti-phase with AMO (r = −0.34, $p < 0.05$). ENSO indices (MEI, SOI) showed significant associations (r = 0.40 and r = −0.40, respectively, $p < 0.05$) and consequently a high influence on the drought conditions inferred with the SPEI for central Chihuahua.

## 4. Discussion

*4.1. Dendrochronological Association*

The combination of several series in a representative regional chronology captures the climatic variability with greater fidelity [47]. This same situation applies when various species are used in the development of a regional series that best represents the annual and interannual climatic variation that characterizes a certain region due to the physiological characteristics of those species [48].

In this study, the integration of a series of chronologies produced a better association with climatic variables. Similar results were obtained for dendroclimatic studies in Chihuahua [49] and at the regional level in northern Mexico [50].

### 4.2. Tree-Ring Chronology and Climatic Variables

The association of the chronology with monthly precipitation indicates that the greatest correlation values occur in the first months of the current growth year, and June, August, September, November, and December of the previous growth year. A situation that was previously corroborated with various conifer species in northern Mexico, where the significant association corresponds to the seasonal period January–August, January–July, and October–July [51]. In this study, the highest association between the regional earlywood chronology and rainfall was obtained for the accumulated precipitation from November of the previous year to June of the current year of growth (r = 0.82), which indicates that the accumulated cool precipitation has the greatest influence on tree growth.

This seasonal period is linked to the ENSO in its warm phase in the Western Sierra Madre, where winter rainfall constitutes a quarter of the annual total, presenting high inter-annual and multi-year variability [17].

The relevance of the winter–spring precipitation in the radial increase in conifers in northern and central Mexico was corroborated by spatial correlation analysis (Figure S1a), as well as also being identified in other research [27–52]. Northern and central Mexico have shown to have spatial correlation values above 0.6, behavior that provides information on the hydroclimatic variability, that characterizes this region and that is influenced by general circulation modes [17].

Summer precipitation, which constitutes > 70 percent of the annual runoff volume, is influenced by the North American Monsoon System and is reflected in the growth of latewood [53]. In northern Mexico, this precipitation is not significantly reflected in the latewood, which is attributed to the fact that much of the rain is lost due to runoff and occurs in a period when the annual radial growth is in its terminal phase. However, this precipitation is important because it is the most important source of water for reservoirs and to refill the water holding capacity of the soil that can be used in the next growing season [54]. The association between monthly and seasonal temperature with ring-width indices varies depending on the geographical location, evapotranspiration rates, and moisture availability for tree growth [55].

Associations between annual radial growth and temperature (mean, minimum, and maximum temperature) were obtained for tree species growing in different mountain ranges of Mexico [56]. In this study, the association between mean temperature and earlywood index was significant but negative, suggesting that increases in temperature accelerated the evaporative process, depleting the soil moisture in a shorter period and affecting the radial growth of the conifer species; similar results were reported by [57]. This negative relationship was also detected in dendroclimatic studies for Mexico [56], and corroborated by spatial-temporal examination with a high association in northern and central Mexico (Figure S1b). Results that explain why high temperatures favor the increase in evapotranspiration and alter the process of photosynthesis [58]—particularly, if we consider that conifer species in this study have a Holarctic origin, with adaptations to low temperatures [59]. This relationship is significant on a global scale because the temperature is an important global component in climate change, presenting modulations in its behavior; its impact seems to be more evident in semiarid regions such as northern Mexico where climate models forecast short- and medium-term increases in temperature and reductions in precipitation [1].

### 4.3. SPEI and Relationship with Climate

We found a significant correlation between the earlywood indices and the annual SPEI, which confirms the potential to develop a drought reconstruction based on tree rings for northern Mexico. This type of analysis helps to understand the variability of drought

conditions in a region (annually or seasonally) because the SPEI considers the precipitation and evapotranspiration derived from temperature [4]. The reconstructed SPEI values were negatively correlated with the mean temperature (r = $-0.68$, $n = 39$, $p < 0.05$), which verifies the close relationship found between this index and the increasing trend in temperature of this region [4]. Similarly, it suggests that global warming increases the intensity of droughts due to an increase in the rate of evapotranspiration [11].

Studies of the SPEI in northern Mexico are very limited; the most recent was developed in central Mexico, specifically in the state of San Luis Potosí [60]. These studies are important and provide technical information about the high- and low-frequency behavior of climatic variability on basins of northern Mexico. In addition, to determine potential trends and the impact of atmospheric circulatory modes in determining water availability [61].

The SPEI values range from $-1.0$ to $1.0$, where $-1.0$ represents the most intense event [8]. In this study, values of $-1.0$, or close to this value were common along the 243 years of reconstructed SPEI (Figure 5). Years or periods with negative SPEI values reduced crop yields, increased livestock death, increased epidemics, and triggered revolts, which in turn affected general socio-economic activities in the state of Chihuahua [62]. Droughts that took place during 1798–1799, 1801–1805, 1809, 1857, 1860, 1862, and 1892–1894 were reported in previous studies occurring simultaneously in northern and central Mexico [48]. This is an indication that they were influenced by large-scale circulatory modes [63]. The wet events of 1785, 1828, and 1852 found in the SPEI reconstruction coincide with seasonal precipitation and streamflow reconstructions in Chihuahua, with a homogenous impact throughout northern Mexico [51]. Extreme droughts were also reported for the southwestern United States in the years 1770, 1820, and 1882–1905 [64], which coincide with negative values in the present SPEI reconstruction.

Ortega-Gaucín's study, [65], characterized the hydrological droughts for the Bravo River Basin and reported an extraordinary drought event from 1997 to 2005, while other studies indicated the presence of an extraordinary drought in the 1950s, which extended to the southwestern United States [18]. In their study of the reconstruction of fire regimes for Durango, [66] presented evidence of wildfires in 1945, 1953, 1962, 1969, 1972, 1988, 1996, 2009, and 2012, which coincide with the events of the last 20 years of the present study, with reconstructed SPEI values below the mean and the increase in Bark beetle outbreaks during drought [67]. Corroborated with low SPEI values in central Chihuahua occurred in 1980, 1996, 2002–2003, 2006, 2011–2012, and 2014, where an increase in temperature was observed in this study. We found a significant correlation between the reconstructed SPEI and precipitation (r = 0.54, $n = 39$, $p < 0.05$), a result that validates the correlation between precipitation amount and droughts in the study area [68].

Multi-taper method spectral analysis and wavelet analysis of the reconstructed SPEI indicated the presence of significant peaks at 2.1 years (a frequency that corresponds to the ENSO signal in its cold phase [La Niña]), which supports the presence of droughts in northern Mexico [69].

The reconstructed SPEI had a positive and significant association with the PDO (r = 0.46, $n = 61$, $p < 0.05$), which may demonstrate the relationship of this circulatory phenomenon with the climatic conditions that could influence the variability of precipitation and evapotranspiration rates in northern Mexico [70]. This relationship was demonstrated through the power spectral density analysis with a significant peak of 55 years; the frequency of 15–70 years is often associated with the PDO signal (Figure S3) [46]. The influence of the AMO showed an opposite behavior (anti-phase) with the SPEI reconstruction (r = $-0.34$, $n = 61$, $p < 0.05$) and a similar situation occurred in northwestern Mexico [52]. Drought events that occurred in the 1930s are attributed to the positive phase of the AMO (periods below the reconstructed historical mean from 1934 to 1940). This behavior is attributed to the presence of high pressure in the area which weakens the low-level jet stream in the Gulf of Mexico [2]. This inverse association between the PDO and the AMO explains the hydroclimatic variability in northern Mexico [63]. Thus, when the PDO is in a positive phase, the AMO is in a negative phase, resulting in wet episodes; however, when

they are opposite (PDO negative and AMO positive), drought episodes may prevail in this region [71]. This is evidenced in the dry years of 1983 and 1984 (SPEI of $-0.89$ and $-0.73$, respectively) that were identified in this reconstruction.

## 5. Conclusions

We identified significant periods of droughts of higher intensity and duration than those documented in historical archives and instrumental records in central Chihuahua. The multiscalar nature of the SPEI allowed us to identify a set of accumulated droughts, which denote the historical conditions that characterize this region, where increases in temperature during the last decades favor a rapid decline in water-vapor deficits and a decrease in annual radial growth rates.

The use of modeled data (obtained from the NLDAS-2 model) from the last 40 years is related to the earlywood and can be used to develop a historical reconstruction of climatic variables such as precipitation and temperature and to reconstruct drought indices.

The PDO and AMO showed a high association with the reconstructed SPEI values in the period 1950–2010; the association with ENSO indices (SOI and MEI) was lower. These results provide a better understanding of the influence of these phenomena on the SPEI variables.

**Supplementary Materials:** The following supporting information can be downloaded at: https://www.mdpi.com/article/10.3390/f13060921/s1, Figure S1. (a) Correlation between regional chronology and average monthly precipitation from November of the previous year to June of the current growth year; (b) and with average monthly temperature from January to July of the current growth year. The green star on both maps represents the study area; Figure S2. Spatial exploration between the regional chronology and the annual SPEI. The green star on map represents the study area; Figure S3. (a) Power spectral density analysis of the reconstructed SPEI; and (b) wavelet analysis; the black out-lines on the cone of influence are significant ($p < 0.05$).

**Author Contributions:** Conceptualization, A.R.M.-S. and J.V.-D.; methodology, A.R.M.-S. and J.V.-D.; formal analysis, A.R.M.-S., L.U.C.-E. and T.C.-A.; investigation, A.R.M.-S. and R.T.-C.; writing—original draft preparation, A.R.M.-S. and J.V.-D.; writing—review and editing, J.E.-Á. and R.T.-C.; project administration, J.V.-D.; funding acquisition, J.V.-D. All authors have read and agreed to the published version of the manuscript.

**Funding:** This research was funded by the Secretary of Public Education-National Council of Science and Technology (CONACYT). "Red dendrocronológica Mexicana: aplicaciones hidroclimáticas y ecológicas" (Mexican dendrochronological network: ecological and hydroclimatic applications), grant number 283134 and "The APC was funded by INIFAP".

**Conflicts of Interest:** The authors declare no conflict of interest.

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
