# Peer review of "Two Centuries of Drought History in the Center of Chihuahua, Mexico"

_forests, doi:10.3390/f13060921_

Round 1
Reviewer 1 Report
This is a great paper, and important regionally. Please pay attention to the points made in the review. Those in red indicate must address points.

Reviewer 2 Report
Global climate change is producing more frequent and more extreme droughts. To better predict the future drought and its impact on water resources, historical knowledge of droughts generated by assimilated data is highly desired. This study aimed to evaluate the association of precipitation and temperature with data from the North American Land Data Assimilation System to reconstruct drought events in the center of Chihuahua, Mexico using the Standardized Precipitation Evapotranspiration Index inferred with tree rings. The authors found that best association among chronologies was obtained with the earlywood band and accumulated seasonal precipitation from November of the previous year to June of the current year; and for temperature it was from January to July. They reconstructed drought index from 1775 to 2017 (243 years), with seven extreme drought events identified. They found significant correlations between reconstructed drought index and the Pacific Decadal Oscillation, Atlantic Multidecadal Oscillation, Multivariate El Niño, and Southern Oscillation Index.
Overall, I think this is valuable contribution to the current understanding on drought induced from global change. The design and presentation of the current study is well done, I believe it is acceptable for publication in its present form.
One more comment only: makes the figures colored would help the readers without increasing cost, as colored figures in electronic publication does not cost more than black-white.
Author Response
No data attached and no responses, due to no suggested changes